# Menstrual Cycle Matters in Host Attractiveness to Mosquitoes and Topical Repellent Protection

**DOI:** 10.3390/insects16030265

**Published:** 2025-03-03

**Authors:** Mara Moreno-Gómez, Sílvia Abril, Júlia Mayol-Pérez, Ana Manzanares-Sierra

**Affiliations:** 1Henkel Ibérica S.A, Research and Development (R&D) Insect Control Department, Carrer Llacuna 22, 1-1, 08005 Barcelona, Spain; 2Department of Environmental Sciences, University of Girona, Carrer Maria Aurèlia Capmany i Farnès, 69, 17003 Girona, Spain; silvia.abril@udg.edu (S.A.); anagema94@gmail.com (A.M.-S.); 3Acondicionamiento Tarrasense, Carrer de la Innovació 2, 08225 Terrassa, Spain; juliamayolperez@gmail.com

**Keywords:** *Aedes albopictus*, complete protection time, hormonal fluctuations, menstrual cycle, topical repellents

## Abstract

This study investigated how the menstrual cycle affects a person’s attractiveness to mosquitoes and their degree of protection when using mosquito repellents. While many factors influence attractiveness to mosquitoes, it is unclear whether the menstrual cycle matters. We found that people were at greater risk of mosquito bites and received less protection from a topical repellent during ovulation than during menstruation and the luteal phase. By clarifying how an important physiological cycle can influence mosquito behavior, our work sheds light on the factors that influence attractiveness to mosquitoes and repellent protection time. Our findings should help guide the development of better ways to protect people from mosquito bites.

## 1. Introduction

Mosquitoes have a tremendous impact on human populations because of their significant effects on public health [1]. They are vectors for a wide range of infectious diseases, including malaria, dengue fever, Zika, and West Nile. According to the World Health Organization (WHO), mosquito-borne diseases cause more than 700,000 deaths annually and affect billions of people each year [2]. In particular, malaria remains a major global health concern, especially in regions with large mosquito populations. Research suggests that nearly half of the global population is at risk of contracting malaria, with most cases likely to occur in African countries. In 2023, it was estimated that 94% of malaria cases (246 million) and 95% of malaria deaths (569,000) occurred in the WHO African Region [3]. Beyond their direct impacts on human health, mosquito-borne diseases have substantial economic consequences. These diseases can lead to increased healthcare costs and loss of productivity, and they place a strain on already fragile healthcare systems, particularly in low-income countries [4,5,6].

Efforts to control mosquito-borne diseases have focused on developing effective mosquito repellents. Spatial repellents interfere with the host-seeking behavior of mosquitoes: the area under protection becomes less attractive or even repellent to mosquitoes [7,8]. Topical repellents act as deterrents, interfering with the ability of mosquitoes to detect and approach human hosts. They are an essential part of bite prevention strategies in high-income countries.

The chemical compounds found in repellents, such as N,N-diethyl-meta-toluamide (DEET), picaridin, and IR3535, have demonstrated remarkable efficacy in reducing the number of mosquito bites [9,10].

It is intriguing that, regardless of the protection strategy used, certain individuals experience higher frequencies of mosquito bites in both laboratory [11,12,13] and field settings [14]. In recent years, research has increasingly underscored the importance of investigating heterogeneity in host attractiveness to mosquitoes, as the results can shine a light on the factors underpinning host-specific differences in mosquito biting rates as well as the complex dynamics linking human physiology and mosquito behavior [12,15,16,17]. At the same time, the specific contributions and interactions of these factors remain poorly understood [18]. There is a widely held belief in the general public that variation in ABO blood type accounts for differences in attractiveness to mosquitoes. However, experimental studies exploring this hypothesis have yielded conflicting results, which suggest that factors beyond ABO blood type are involved [19,20,21].

Research has examined other potential factors, including genetic background [22,23,24] and visual cues, such as clothing color. Specifically, wearing dark clothing enhances an individual’s visibility against lighter backgrounds, drawing in more mosquitoes [25,26]. In addition, chemical cues, especially odor-mediated signals, are widely known to play a significant role in mosquito host-seeking behavior [27,28,29]. One of the most well-established of these cues is carbon dioxide (CO_2_), a kairomone exhaled by vertebrates and a reliable indicator of host presence for mosquitoes [30]. However, mosquitoes respond to more than CO_2_—they detect other volatile organic compounds (VOCs) emitted by the human body, such as alkanes, aromatic hydrocarbons, alcohols, aldehydes, ketones, amines, and carboxylic acids [31,32]. The abundance and composition of these compounds can vary significantly among individuals, which may be another source of host-specific differences in mosquito biting rates [31].

Human body odor is primarily the result of two processes. The first is the breakdown of sebum and sweat on the skin’s surface as a result of microbial activity, leading to oxidation and odor production [33,34]. The second is the release of VOCs, also known as skin gases; these compounds are emitted by sources within the body and released via the skin [35,36,37]. Some of these VOCs, such as 3-methyl-1-butanol, are known to attract mosquitoes [38], while others, including 6-methyl-5-hepten-2-one, octanal, nonanal, decanal, and geranylacetone, have been found to repel mosquitoes [39,40,41]. Furthermore, an individual’s body odor seems to arise from the presence of specific microorganisms and is correlated with skin microbial profiles [16,42,43]. For example, individuals who are more attractive to mosquitoes tend to have a larger number but lower diversity of skin bacteria [15,43].

Additional factors known to influence host attractiveness to mosquitoes are pregnancy [44,45,46], the use of skin care products [41], and the consumption of specific foods, such as bananas or alcohol [47,48,49]. The precise biological mechanisms underlying these effects remain unclear, and little is known about how body odor may be contributing to these patterns. Factors such as gender [50,51] and age [13,24,52] have also been studied, but the results have largely been inconsistent, likely because of differences in testing methods or mosquito species. As Ellwanger et al., (2021) emphasized, an individual’s likelihood of being bitten by a mosquito is the product of a complex interplay of host-related factors, environmental conditions, and the inherent characteristics of mosquitoes [31].

Given that mosquitoes significantly affect the health and well-being of human populations, it is crucial to better understand the factors influencing host attractiveness if we wish to design effective disease prevention and control strategies. By unraveling the complex interactions at play, we can gain valuable insights that will guide the development of targeted interventions and improve existing methods for repelling mosquitoes.

To date, the menstrual cycle is one important factor that has received limited attention in studies of variability in host attractiveness to mosquitoes. The menstrual cycle is a dynamic physiological process that is characterized by hormone fluctuations. During the menstrual cycle, the complex interactions of hormones such as estrogen and progesterone serve to regulate ovulation and menstruation [53,54]. Estrogen peaks during the follicular phase, reaching its highest levels at ovulation. In contrast, during menstruation at the end of the luteal phase, both estrogen and progesterone levels drop to their lowest. These hormonal fluctuations not only regulate ovulation and menstruation but also are responsible for various physiological and behavioral changes, such as alterations in body temperature [55,56], metabolic processes [57,58], and scent production [59]. Given these hormonal variations, it is reasonable to hypothesize that attractiveness to mosquitoes may vary across the different stages of the menstrual cycle. While few studies have systematically tested this hypothesis, evidence suggests that the hormonal changes occurring during the menstrual cycle could impact an individual’s attractiveness to mosquitoes: indeed, an increase in mosquito bites has been seen when individuals are ovulating [60]. It is thought that mosquitoes are attracted to the estrogen being emitted by the skin during this period, given that amino acids are emitted at a relatively constant rate across the menstrual cycle [60].

We conducted a study exploring how the menstrual cycle affects host attractiveness to mosquitoes utilizing the arm-in-cage test, which is described in detail in the efficacy guidelines published by the European Chemical Agency (ECHA) [61], Environmental Protection Agency (EPA) [62], and WHO [63]. This method provides controlled and standardized conditions for assessing host attractiveness to mosquitoes and the duration of protection afforded by topical mosquito repellents. We divided the menstrual cycle into three distinct phases: menstruation, ovulation, and the luteal phase. Each phase is characterized by unique hormonal patterns and physiological changes. Our objective was to investigate whether the menstrual cycle can influence host attractiveness to mosquitoes and, consequently, the duration of repellent efficacy. Having greater clarity about this relationship should help in the development of strategies for protecting public health worldwide.

## 2. Materials and Methods

This study was conducted in the Henkel Ibérica Research and Development (R&D) Insect Control Department (Spain) between March and June 2022. The work described herein was approved by the ethics committee of Henkel AG & Co. KGaA (Düsseldorf, Germany). It met the company’s corporate standards, which ensure health, safety, and respect for the environment as well as the protection and ethical treatment of all study participants. In brief, we conducted replicated arm-in-cage trials during the different phases of the participants’ menstrual cycles. During these trials, one arm was treated with a topical repellent to measure the duration of repellent efficacy, and the other arm was left untreated to determine control levels of mosquito activity (i.e., landing rate [LR]: number of mosquitoes landing per minute). To avoid altering the natural odors, skin chemistry, or temperature, the skin was not cleaned before the start of the test. For each participant, arm status (treatment vs. control) remained the same across the experiment. Our methodology is described in greater detail below. This study was conducted at 25 ± 5 °C and 50 ± 10% RH.

*Participants*: Five women aged 25–44 were recruited for this study. They signed a written informed consent form, which explained the study’s purpose and procedures, their role and responsibilities as participants, and their right to withdraw from the study at any point. To take part in this study, participants had to have at least one year of data on their menstrual cycles. These data were used in conjunction with the application WomanLog, which tracks and predicts the timing of menstrual cycles, fertility, and ovulation. It has been installed over 20 million times and has more than 1.5 million monthly active users, resulting in reliable forecasting data [64]. Having at least one year of data helped ensure the application had sufficient historical data to yield reliable predictions for each participant. We focused on three phases: ovulation, menstruation, and the luteal phase. The dates of each phase for each participant were determined using the application’s individualized predictions, rather than assuming a fixed-length menstrual cycle.

*Mosquitoes*: We used a strain of *Aedes albopictus* (Skuse 1895) that had been obtained from the Entostudio Test Institute (Italy) in 2013 and that was subsequently reared in-house (conditions: temperature = 25 ± 2 °C, relative humidity = 60 ± 5%, and photoperiod = 12:12 [L:D]). During each trial, 40–45 mosquitoes were released into a 0.040-m^3^ enclosure (i.e., cage). This number of mosquitoes allowed us to achieve the minimum LR specified by ECHA guidelines (i.e., 20 landings/min) [61] as well as the minimum LR required by WHO guidelines (i.e., 10 landings/30 s or 20 landings/min) [63]. The mosquitoes in the cage were replaced with new mosquitoes if the target LR was not achieved during the control trials [61,63]. Only female mosquitoes between 5 and 10 days in age were used. They were not fed any blood. Instead, throughout the trials, they had ad libitum access to a 10% sucrose solution to help ensure they remained in good health.

*Repellent*: The repellent formula was provided by Endura S.p.A. (Bologna, Italy). It did not contain any fragrances, and the co-formulants were alcohol based. The active substance was 15% DEET (CAS number 134-62-3), which was chosen because it is one of the most common chemical insect repellents on the market. It has been in use worldwide since the 1950s [65], and the WHO recommends that it be employed as the positive control when evaluating topical repellents [63]. We used a percentage of DEET that ensured that the repellent would result in complete protection times (CPTs) of less than 8 h, with a view to facilitating comparisons among study participants. The dose was 0.5 g of repellent per 600 cm^2^ of skin surface, a choice that was informed by past work of ours [66].

*Experimental trials*: We conducted arm-in-cage testing with each participant during each of the three phases of the menstrual cycle over the course of three consecutive menstrual cycles, resulting in a total of nine testing periods. During each testing period, we ran trials characterizing repellent efficacy and control levels of mosquito activity. Since the participants were not all synchronized, the tests were carried out on different days. We specifically used a standard sleeved arm-in-cage test (see ECHA guidelines [61]; Figure 1), an approach in which participants wear sleeves that limit the surface area of skin exposed to the mosquitoes, providing greater protection against potential bites. The participants’ sleeves exposed 100 cm^2^ of the underside of their forearms (which has fewer hairs), a surface area chosen in accordance with ECHA recommendations [61]. Each study participant had her own set of sleeves—one for the arm treated with repellent and one for the arm left untreated. In addition, the participants always wore gloves to protect their hands. As per EU guidelines, participants were asked to avoid the use of nicotine, alcohol, fragrances (e.g., perfumes, body lotions, soap), and repellents for 12 h prior to and during all testing periods. They were also instructed to maintain a medium-low level of physical activity prior to and during the trials to avoid any potential changes in body temperature that could influence the results.

### 2.1. Measuring Repellent Efficacy

During each of the testing periods for each participant, we estimated the repellent’s CPT, which is defined as the period over which the repellent’s level of protection is 100%. We performed four consecutive replicates, each using a different cage, resulting in a total of 60 estimates of CPT for each menstrual phase (4 replicates × 3 menstrual cycles × 5 participants) and 180 estimates of CPT overall (60 × 3 menstrual phases). The testing period lasted between one and two days, depending on how quickly CPT ended for each replicate.

In these trials, participants first applied repellent evenly across one of their forearms using a pipette. The amount of repellent to be applied was calculated based on the dose mentioned above and the surface exposed by the sleeve (i.e., 100 cm^2^). During repellent application, the product was applied to an area slightly larger than the area to be exposed (i.e., there was overlap between the area treated with repellent and the area covered by the sleeve), as per ECHA guidelines [61].

Next, once per hour and under the supervision of a trained researcher, the participants introduced their forearms into the cages for a 3-min exposure period. This process continued for a maximum of 8 h, or until the level of protection dropped below 100%, whichever occurred first. We followed European guidelines for estimating CPT, which are based on mosquito probing (i.e., when a mosquito penetrates the skin with its mouthparts without ingesting any blood) [61]. These guidelines indicate that once the first instance of probing is observed, it must be validated by a second instance of probing that happens during the same or the following 3-min exposure period. Then, the exposure period that occurred prior to the first probing event is identified, and CPT is the amount of time between repellent application and this preceding exposure period. Any probing mosquitoes were promptly dispelled by the participant shaking their arm.

### 2.2. Measuring Control Levels of Mosquito Activity

During each of the testing periods for each participant, we also characterized the level of mosquito activity in the absence of the repellent. These trials were conducted at the beginning and end of each replicate, as well as every two hours throughout the replicate. In this case, however, the participants introduced their untreated forearms into the cages for a 3-min exposure period, and the number of mosquito landings was recorded. A landing occurs when a flying mosquito settles on the skin without biting or probing. Any landing mosquitoes were promptly dispelled by the participant shaking their arm. Using these data, we ensured that the mosquitoes maintained sufficient levels of activity across the testing period [61,63] and we were able to estimate LR.

Thus, because CPT varied across replicates, so did the number of LR estimates (range: 2–4).

### 2.3. Measuring Temperature

Participant body temperature was measured at two locations—the forehead and wrist—at the beginning and end of each testing period. We used a handheld infrared thermometer (DT-8809C, Pioway Medical Lab Equipment Co., Ltd. Nanjing, China), which is designed for taking contact-free temperature measurements. The thermometer has a precision of ±0.3 °C when body temperature is between 35.0 and 42.0 °C and environmental temperature is between 10 and 40 °C; conditions during the study fell within these windows. It is important to note that, at the beginning of the study, temperature data were not recorded for four participants during menstruation and one participant during ovulation. Thus, for each location (forehead and wrist), we obtained 28 measurements during ovulation, 20 measurements during menstruation, and 30 measurements during the luteal phase (total: 78).

## 3. Statistical Analysis

R was used to perform all the statistical analyses [67], for which the alpha level was always 0.05.

The median CPT values and corresponding 95% confidence intervals (CIs) were estimated using Kaplan–Meier survival analysis in accordance with WHO guidelines [63]. A mixed-effects Cox proportional hazards model was used to analyze the effect of menstrual cycle phase (ovulation, menstruation, or luteal phase) on CPT (*n* = 180). The model included the menstrual cycle phase as a fixed effect and participant identity as a random effect, to account for the repeated measures.

A generalized linear mixed model (GLMM) was used to assess differences in LR among menstrual cycle phases. The model utilized a Poisson error distribution (identity link function) and was performed using the glmmPQL function in the MASS package. The response variable was LR (*n* = 560), and the fixed effect was menstrual cycle phase (ovulation, menstruation, or luteal phase); participant identity was a random effect.

To evaluate differences in forehead and wrist temperature measurements among menstrual cycle phases, GLMMs with a Gamma distribution (log link function) were performed using the glmTMB package. The response variable was temperature at a given location (*n* = 78), and the fixed effect was menstrual cycle phase (ovulation, menstruation, or luteal phase); participant identity was a random effect.

## 4. Results

Median CPT (±95% CI) was significantly influenced by menstrual cycle phase (luteal phase: 5.00 h [±0.19] > menstruation: 4.00 h [±0.34] > ovulation: 4.00 h [±0.79]) (Figure 2).

The mixed-effects Cox proportional hazards model indicated that CPT was significantly influenced by participant identity. The standard deviation (SD) for the latter’s estimator was 0.5229, and the variance was 0.2734. This result suggests that differences among participants significantly contributed to variability in CPT.

The Cox model also indicated that menstrual cycle phase had a significant influence on CPT (Table 1; all *p*-values < 0.05). Based on the hazard ratios (HRs), we can see that, during ovulation, the risk of losing complete protection against mosquitoes was 84.76% higher than during menstruation and 438.18% higher than during the luteal phase (Table 1). This result implies that CPT was shorter during ovulation than during menstruation or the luteal phase (Figure 2). During menstruation, the risk of losing complete protection against mosquitoes was 45.88% lower than during ovulation but 191.29% higher than during the luteal phase. Finally, during the luteal phase, the risk of losing complete protection against mosquitoes was 65.67% lower than during menstruation and 81.42% lower than during ovulation. Overall, these results underscore that CPT was longest during the luteal phase and shortest during ovulation (Table 1; Figure 2).

Menstrual cycle phase also influenced mean LR (±SD) (ovulation: 99.05 [±37.24], menstruation: 90.57 [±37.57], and luteal phase: 87.09 [±37.28]). LR was significantly higher during ovulation compared to during menstruation (GLMM: t_552_ = 2.18, *p* = 0.029) and to during the luteal phase (GLMM: t_552_ = 3.28, *p* = 0.001). No statistical differences were seen in LR during menstruation versus the luteal phase (GLMM: t_552_ = 1.09, *p* = 0.273) (Figure 3).

Menstrual cycle phase had no influence on mean forehead temperature (±SD) (ovulation: 36.51 °C [±0.33], menstruation: 36.49 °C [±0.40], and luteal phase: 36.44 °C [±0.33]; GLMM: χ^2^ = 2771, df = 2, *p* = 0.87), nor did it have an influence on mean wrist temperature (±SD) (ovulation: 35.67 °C [±0.50], menstruation: 35.75 °C [±0.51], and luteal phase: 35.71 °C [±0.45]; GLMM: χ^2^ = 0.16, df = 2, *p* = 0.92).

## 5. Discussion

Our research addresses a current gap in knowledge regarding the menstrual cycle’s influence on host attractiveness to mosquitoes and the efficacy of topical repellents. Our study reveals that there may indeed be a relationship. Specifically, we observed that ovulation was associated with the shortest complete protection time and the highest mosquito landing rate. In contrast, the luteal phase was associated with the longest complete protection time and the lowest mosquito landing rate (although landing rate was equivalent during menstruation). These patterns may be linked to hormone fluctuations during the menstrual cycle, which can influence body odor, temperature, and/or skin chemistry, factors that may increase a host’s attractiveness to mosquitoes. This increase in attractiveness is likely linked to the decrease in complete protection time: during ovulation, the risk of losing protection against mosquitoes was 1.84 times higher than during menstruation and 5.38 times higher than during the luteal phase.

A growing body of evidence suggests that female body odor changes across the menstrual cycle, with men perceiving odors occurring during ovulation as more attractive. Ovulation is a menstrual cycle phase during which fertility and estrogen levels are high. Men rate these odors as more appealing than those occurring during the low-fertility phase of the cycle, when levels of progesterone are higher [68,69,70]. Although our study focuses on how the menstrual cycle affects attractiveness to mosquitoes rather than the attractiveness of female body odor to other humans, these findings are still relevant. They provide evidence that fluctuations in female physiology can alter odor profiles, which may influence responses by humans and other species such as mosquitoes.

Our study focused exclusively on understanding how attractiveness to mosquitoes and, consequently, complete protection time were influenced by each phase of the menstrual cycle. That said, we did not characterize hormone levels or skin chemistry, information that could have provided further insight into the mechanisms underlying our observations. Below, we explore potential explanations for our findings and emphasize the need for further research in this complex area.

Several mechanisms could explain the shorter protection time and increased attractiveness to mosquitoes during ovulation. During this phase of the menstrual cycle, estrogen levels peak [71], triggering the release of a mature egg and increasing the emission of volatile compounds such as lactic acid and pheromones, which have been shown to attract mosquitoes [60]. These dynamics could help explain the higher mosquito landing rate we observed during this phase. In contrast, during menstruation, estrogen and progesterone levels are at their lowest [71], likely resulting in the emission of fewer of the volatile compounds that attract mosquitoes, which might have contributed to the lower landing rate we observed during this phase. Finally, during the luteal phase, progesterone levels climb and estrogen levels drop, which might lead to an odor profile even less attractive to mosquitoes [71].

The rise in body temperature during ovulation may further enhance host attractiveness to mosquitoes [69]. Estrogen plays a role in regulating body temperature, and during ovulation, it can cause slight increases in body temperature [72]. Mosquitoes are highly sensitive to heat and perspiration, and these physiological changes could make hosts more attractive, even when repellents are used [73]. A rise in body temperature could also accelerate the evaporation of repellents, reducing their efficacy and shortening complete protection time. Although it is well known that temperature varies across the menstrual cycle [74,75,76], we did not see any significant differences in forehead and wrist temperatures. It may be that our thermometer was not sufficiently precise; Sumic and Ravlic (2013) suggested that temperature differences among menstrual cycle phases may differ by as little as a few decimal points [77]. Additionally, temperature location might matter. Previous research has indicated that temperature readings may vary depending on the measurement site: rectal temperatures most accurately reflect core body temperature and are generally higher than temperature measurements obtained from the mouth, ear, or underarm [75,78].

The interaction between repellents, hormonal changes, and skin physiology is another angle that should be considered. The skin is the largest organ in the body, and both the dermis and epidermis contain estrogen receptors and, to a lesser extent, progesterone receptors [75]. Fluctuations in these hormones, particularly those of a cyclic nature, influence various skin characteristics, including lipid secretion, sebum production, skin thickness, fat deposition, hydration, and barrier functions [76,78]. Research has shown that lipid secretion by the skin is much higher during the luteal phase of the menstrual cycle [79]. High levels of estrogen suppress sebum production, and the sebum content of the skin is therefore lowest during ovulation [80]. In addition, dermis thickness, which is correlated with collagen content, is influenced by estradiol [81,82]. Hall and Phillips (2005) observed a 30% increase in dermal thickness in women taking estrogen replacement therapy [82]. Eisenbeiss et al., (1998) demonstrated that skin thickness varies across the menstrual cycle: it is thinnest during menstruation, when estrogen and progesterone levels are low; it grows thicker as estrogen levels rise during ovulation; and it is thickest during the luteal phase [83]. The same study found that skin echodensity (i.e., a metric reflecting skin density) increased slightly from menstruation to ovulation but then decreased during the luteal phase; however, these changes were not statistically significant [83]. Absolute skin thickness varied among locations but universally demonstrated a hormone-related increase around the time of ovulation. Estrogen also induces fat accumulation in subcutaneous tissues [84], and subcutaneous fat associated with the thighs and abdomen reached maximum thickness during menstruation and minimum thickness during the luteal phase [85]. Few studies have specifically focused on how the menstrual cycle affects skin hydration, but Berardesca et al., (1989) found that, while menstrual cycle phase had no influence on skin hydration or surface water loss along the volar forearm and upper thigh, levels of hydration and surface water loss were slightly higher on day 25 of the cycle (during the luteal phase) than on day 10 (closer to ovulation [86]). However, this study only collected measurements at two time points.

Given that hormones can clearly alter skin characteristics, it seems plausible that the decrease in repellent efficacy that we observed during ovulation could partially be explained by these physiological changes. The lower sebum levels and increased skin thickness associated with ovulation could influence how topical repellents are absorbed or retained. However, further research is needed to fully understand the relative contributions of these physiological changes, as well as their impacts on the production of volatile compounds and skin odor. Such work will be crucial in identifying the factors that play the most significant role in altering repellent efficacy and host attractiveness to mosquitoes during the menstrual cycle.

We used DEET in this study because it is the oldest and the most powerful topical repellent available on the market; it is thus the standard of reference [87]. However, other compounds, such as picaridin, N,N-diethyl phenylacetamide (DEPA), IR3535, and plant-based alternatives, could respond differently to shifts in skin chemistry and hormone levels, potentially yielding other patterns of efficacy [87]. Future research should also explore whether the menstrual cycle influences the performance of other compounds in the same way. It could also examine the effects of menopause and perimenopause on repellent efficacy. The relationship between hormone levels and repellent efficacy might also be influenced by the changes in the skin microbiome over the course of the human lifespan, as well as by decreases in sebum production after menopause [88].

It is also known that different mosquito species display different responses to various factors affecting host attractiveness [27,89]. It would be useful to expand on the findings of this study by conducting research using additional mosquito genera, such as *Culex* and *Anopheles*. These species exhibit distinct behavioral and ecological traits, and their inclusion could provide further insight into how the menstrual cycle and hormone fluctuations influence host attractiveness to different mosquito species. Obtaining results for more genera would not only enhance the broader applicability of this research, but also provide a more comprehensive understanding of the complex interactions among repellents, skin physiology, and mosquito behavior.

Finally, while our study had a small sample size (five participants), we have previously found that there is as much variability within participants as among participants. This discovery suggests that valuable insights can be obtained by collecting repeated measures from even a small set of individuals [17]. We acknowledge that the small number of participants may limit the broader applicability of our findings; however, we believe that our results, although preliminary in nature, could provide a starting point for larger studies.

Understanding how the menstrual cycle affects host attractiveness to mosquitoes is scientifically intriguing and also has practical implications. We uncovered significant differences in mosquito landing rate and repellent efficacy across the course of the menstrual cycle, and these findings shed light on the complex interplay between hormone fluctuations and mosquito behavior. If hormone fluctuations do indeed impact host attractiveness to mosquitoes, it would make sense to explore the development of personalized strategies for preventing mosquito bites. Tailoring protection measures to account for an individual’s hormonal status could enhance the effectiveness of mosquito repellents and reduce the risk of mosquito-borne diseases.

## Figures and Tables

**Figure 1 insects-16-00265-f001:**
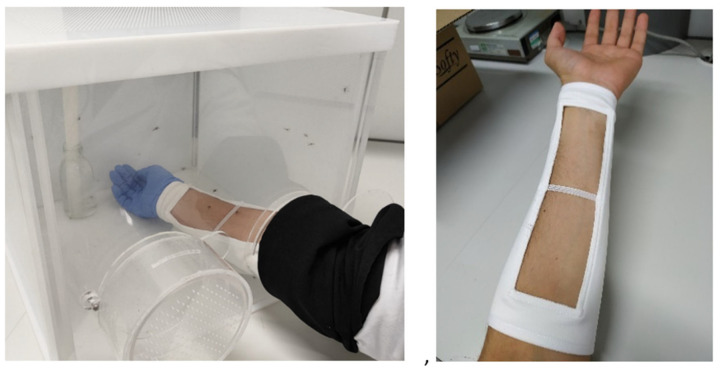
Sleeved arm-in-cage test (0.040-m^3^) where 100 cm^3^ of skin was exposed.

**Figure 2 insects-16-00265-f002:**
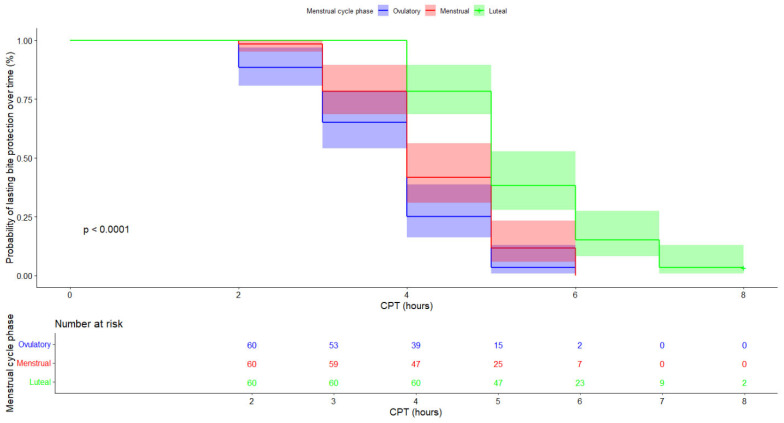
Kaplan–Meier curves depicting patterns of mosquito protection time (CPT) for the three menstrual cycle phases (ovulation, menstruation, and the luteal phase). The table underneath the plot shows the number of individuals at risk of losing complete protection against mosquitoes at each time point for each menstrual cycle phase. There was a significant difference in CPT among menstrual cycle phases (*p* < 0.0001). Complete protection was retained for longer during the luteal phase than during menstruation and ovulation.

**Figure 3 insects-16-00265-f003:**
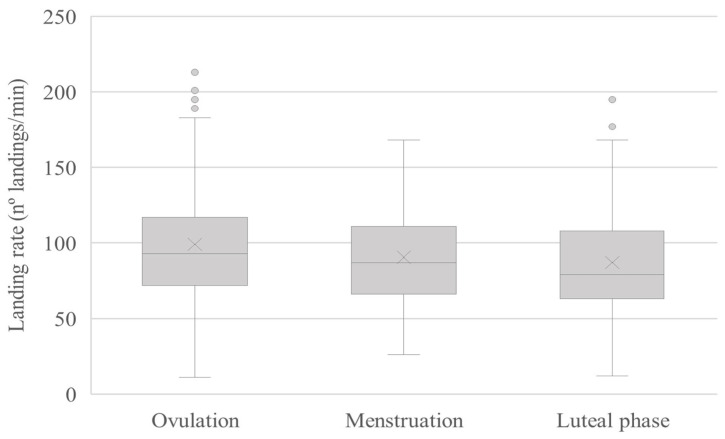
Landing rates during ovulation, menstruation, and the luteal phase of the menstrual cycle. Grey points are outliers.

**Table 1 insects-16-00265-t001:** Coefficients and hazard ratios (HRs) from the mixed-effects Cox proportional hazards model analyzing the influence of menstrual cycle phase on complete protection time against mosquitoes.

Reference Category ^1^	Comparative Category	Cox Coefficient (β) ^2^	HR ^3^	95% CI for HR	z-Score	*p*-Value	% Change in Risk ^4^
Ovulation	Menstruation	−0.614	0.541	0.37–0.78	−3.24	<0.01	−45.88%
	Luteal phase	−1.683	0.186	0.12–0.29	−7.65	<0.0001	−81.42%
Menstruation	Ovulation	0.614	1.848	1.27–2.68	3.24	<0.01	+84.76%
	Luteal phase	−1.069	0.343	0.23–0.51	−5.19	<0.0001	−65.67%
Luteal phase	Ovulation	1.683	5.382	3.49–8.28	7.65	<0.0001	+438.18%
	Menstruation	1.069	2.913	1.94–4.36	5.19	<0.0001	+191.29%

^1^ Reference category: the baseline category against which the HRs for other categories are compared (HR = 1 for the baseline category). ^2^ Cox coefficient (β): statistic expressing the influence of menstrual cycle phase on CPT. A positive coefficient indicates that the risk of losing complete protection against mosquitoes was higher, compared to the reference category, while a negative coefficient indicates that this risk was lower. ^3^ Hazard ratio: statistic expressing the relative risk of losing complete protection against mosquitoes for a given category compared to the reference category. Values exceeding 1 indicate that the relative risk was greater, while values smaller than 1 indicate that the relative risk was lower. ^4^ Change in risk (%): (HR − 1) × 100. This figure indicates the percent increase or decrease in the risk of losing complete protection against mosquitoes relative to the reference category. Positive values indicate a relatively higher risk, while negative values indicate a relatively lower risk.

## Data Availability

The datasets generated during and/or analyzed during this study are available from the corresponding author upon reasonable request.

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
