# Peer review of "Menstrual Cycle Matters in Host Attractiveness to Mosquitoes and Topical Repellent Protection"

_insects, 2025, doi:10.3390/insects16030265_

Round 1
Reviewer 1 Report
Comments and Suggestions for Authors
I would like to congratulate the authors on this original, interesting and applied work and also for the clarity with which the text has been written. There are still many unknows regarding the interpersonal and even intrapersonal variability of mosquito attraction, but we need to start somewhere and without a doubt, the hormonal effect of the menstrual cycle deserves attention, and more for personal protection measures in mosquito-borne diseases contexts.
The results will be very useful for interpreting certain research and even for the evaluation of repellents and recommendations for the application.
What a pity you did not include menopausal women in your study. It would have been interesting to know also the results in this part of the population, as hormonal changes greatly affect the skin and therefore may affect the efficiency of topical repellents.
I have made some comments in the document, some of the issues are well answered in the discussion, but in my opinion, some basic information on the menstrual cycle itself and its impact on other parameters would help the reader to better understand the entire paper.

Reviewer 2 Report
Comments and Suggestions for Authors
Comments:
- Lines 35–36: I recommend listing the keywords in alphabetical order.
- Lines 41–43: I find the statement misleading and inflated, as current WHO data does not support claims of millions of deaths or billions of people affected annually by mosquito-borne diseases.
- The study’s use of only five participants is very limiting, and the tests were conducted in a small laboratory chamber. The dimensions of the chamber should be provided. Moreover, it is known that the repellent effect of DEET (15%) lasts only a few hours in individuals with varying characteristics. It would be beneficial to include results from studies conducted with different age groups and genders. Therefore, the manuscript should clearly state that tests conducted under field or semi-field conditions would be more realistic. Additionally, presenting the average repellent values along with their standard error would improve the results.
- In the figure title on line 194, “100 cm³” should be corrected to “100 cm².”
- On line 237, it is mentioned that the device measuring body temperature records values between 35.0 and 42.0°C. Is a temperature above 38°C not indicative of illness, etc.? Temperatures between 40–42°C are very concerning for humans. Could there be an issue with the device’s measurement?
- All references should be reviewed, and scientific names within the text should be italicized.
Author Response
Please, see the attachment
